# Determination of Partial Propagation Velocity and Partial Isentropic Compressibility Coefficient in Water–Ethanol System

**DOI:** 10.3390/s24134061

**Published:** 2024-06-21

**Authors:** Carlos A. B. Reyna, Ediguer E. Franco, Jose H. Lopes, Marcos S. G. Tsuzuki, Flávio Buiochi

**Affiliations:** 1Department of Mechatronics and Mechanical Systems Engineering, Polytechnic School Sao Paulo University, Sao Paulo 05508-030, Brazil; jose.andrade@arapiraca.ufal.br (J.H.L.); mtsuzuki@usp.br (M.S.G.T.); fbuiochi@usp.br (F.B.); 2Facultad de Ingeniería, Universidad Autónoma de Occidente, Cali 760030, Colombia; 3Arapiraca Campus, Faculty of Physics, Federal University of Alagoas, Arapiraca 57309-005, Brazil

**Keywords:** propagation velocity, layered models, water–ethanol, partial properties

## Abstract

This study introduces an innovative approach to the layered model, emphasizing the physical–chemical characterization of miscible liquid systems through ultrasonic techniques, with a specific focus on the water–ethanol system used in pharmaceutical formulations. Traditional characterization methods, while effective, face challenges due to the complex nature of solutions, such as the need for large pressure variations and strict temperature control. The proposed approach integrates partial molar volumes and partial propagation velocity functions into the layered model, enabling a nuanced understanding of miscibility and interactions. Ultrasonic techniques are used to calculate the isentropic compressibility coefficient for each component of the mixture as well as the total value using an additive mixing rule. Unlike conventional methods, this technique uses tabulated and experimental data to estimate the propagation velocity in the mixture, leading to a more precise computation of the isentropic compressibility coefficient. The results indicate a significant improvement in predicting the behavior of the water–ethanol system compared to the classical layered model. The methodology demonstrates the potential to provide new physicochemical insights that can be applied to other miscible systems beyond water–ethanol. This research has implications for improving the efficiency and accuracy of liquid medication formulations in the pharmaceutical industry.

## 1. Introduction

The physical–chemical characterization procedure for binary miscible solutions is well established. Using the density of the experimental mixture as an input, it is possible to determine the molar and partial molar properties of the system. An example of this type of binary solution is the water–ethanol system, whose characterization is widely used in pharmacology for the development of liquid medications. In this case, the characterization may be used to overcome the low solubility of the active components in water [1].

Ultrasound characterization techniques have great potential for industrial applications. Despite that, the literature shows some tests that were not very successful. Some studies reported the use of an ultrasonic technique to monitor the alcoholic fermentation process online. In these cases, different sources of carbohydrates were used to obtain fermented drinks such as beer, wines [2] and other miscible ternary systems [3,4], but the theoretical comparison was not very accurate. Other studies attempted to show the relationship between chemical and acoustic properties (such as propagation velocity and attenuation), in some concentration ranges of miscible binary [5] and ternary mixtures, but it was not possible to establish a correlation between propagation velocity and concentration [3,6].

The Newton–Laplace equation was used [7,8,9] to calculate the isentropic compressibility coefficient $K_{s}$ in water–ethanol systems considering the properties of the mixture. The authors demonstrate that the ultrasonic technique is a good alternative tool for the characterization of this kind of mixture. Next, refs. [8,9] propose a molecular explanation for the parabolic behavior of the isentropic compressibility. It should be noted that $K_{s}$ is difficult to measure with conventional methods, requiring large pressure variations (in liquids) while keeping the temperature constant [10,11].

The layered model is useful for predicting the behavior of nonmiscible liquid mixtures such as suspensions or emulsions [12,13,14,15,16]. However, the interpretation of a solution composed of partial molar volumes configured as layers for each species allows the implementation of the layers equation in miscible systems (solutions). A similar approach was proposed [17], but it takes into account the molar volume in the pure state for each substance instead of the partial molar volume in the mixture commonly used in miscible systems.

In this work, the propagation velocity functions for each substance in the solution were calculated using the partial molar properties in the layered model. Those results were used to determine the isentropic compressibility coefficient for each species in the water–ethanol system. Finally, the total value of $K_{s}$ was estimated using an additive mixing rule, to be compared to those calculated conventionally.

## 2. Theoretical Background

The layered model proposes that the total propagation time of an ultrasonic wave through a homogeneous mixture is the sum of the flight time in each substance, considering their volumetric proportions as if they were nonmiscible layers [13,14,15,18] (see Figure 1):(1)$$t_{m}=t_{a}+t_{b}=\frac{X_{a}}{c_{a}}+\frac{X_{b}}{c_{b}}=\frac{X_{m}}{c_{m}},$$
where *t* is the ultrasonic time of flight, *X* is the wave path length, *c* is the propagation velocity and the subscripts *m*, *a* and *b* refer to the mixture and the chemical species *a* and *b*, respectively.

Figure 1 shows the arrangement of two transducers that are placed apart by a known distance ($X_{m}$). The first transducer operates as an emitter/receiver (Tx), and the second one operates as a receiver/reflector (Rx). The excitation of the transducer Tx generates the ultrasound wave that propagates through the sample until part of its energy is received by the transducer Rx and part of it is reflected back to the transducer Tx. At this moment, the signal s(t) is detected by the transducer Rx. The ultrasound wave that returns backward is detected by the transducer Tx as signal q(t). As the distance $X_{m}$ is known, the delay $t_{m}$ (time of flight) between the signals s(t) and q(t) allows for determining the propagation velocity $c_{m}$ in the mixture. The same measurement scheme was previously used by us in applications to determine the water content in water-in-crude oil emulsions and to analyze saline solutions [14]. Multiplying the two terms of Equation (Equation 1) by a constant cross-sectional area gives:(2)$$\frac{v_{m}}{c_{m}}=\frac{v_{a}}{c_{a}}+\frac{v_{b}}{c_{b}},$$
where *v* denotes spatial volume and $c_{a}$ and $c_{b}$ are the usual inputs to the layered model (Figure 1b bottom). The relationship between the propagation velocity of the mixture $c_{m}$ and the volume fraction $\phi_{a}$ ($\frac{v_{a}}{v_{m}}$) for a binary mixture is:(3)$$c_{m}=\frac{1}{\frac{\phi_{a}}{c_{a}}+\frac{1-\phi_{a}}{c_{b}}}.$$
where the volume fraction $\phi_{a}$ is defined as the volume $v_{a}$ of the pure substance *a* divided by the total volume of the mixture $v_{m}=v_{a}+v_{b}$.

Equation (Equation 3) is useful for the correlation between the propagation velocity and concentration of nonmiscible systems, where the volume of each substance remains the same after the mixing process without changing the combined volume [14,19].

However, volume is not a conservative property for miscible systems. In this case, the variability of the volume mixture in relation to the initial volumes of the pure substances is associated with molecular polar interactions (hydrogen bonds) between species [1,8]. The molar volume $V_{m}$ of the solution can be calculated from the density $\rho_{m}$ measured experimentally from the mixture, the molecular mass $M_{i}$ and the molar fraction $x_{i}=\frac{n_{i}}{n_{m}}$, where $n_{i}$ is the mole amount of a pure substance (i=a,b) and $n_{m}$ is the total mole amount of all substances in the mixture, as follows [1,20]:(4)$$V_{m}=\frac{x_{a}M_{a}+x_{b}M_{b}}{\rho_{m}}.$$

The experimental molar volume $V_{m}$ as a function of the molar concentration $x_{i}$ can be adjusted by a third-degree polynomial equation (Equation ([Disp-formula FD5a-sensors-24-04061])). The choice of a third-degree polynomial is the most suitable for increasing the correlation coefficient $R^{2}$ with the experimental data [1]. Applying the Legendre Transformation, a well-established mathematical operator to handle thermodynamic properties in Equation ([Disp-formula FD5a-sensors-24-04061]), a new couple of variables is obtained, whose physical interpretation is the partial molar volumes, ${\bar{V}}_{a}$ in Equation () and ${\bar{V}}_{b}$ at Equation ([Disp-formula FD5b-sensors-24-04061]) [20,21]. This couple of partial molar volumes fulfills an additive rule.
(5a)$$\begin{array}{c} V_{m}=Ax_{a}^{3}+Bx_{a}^{2}+Cx_{a}+D, \end{array}$$
(5b)$$\begin{array}{c} {\bar{V}}_{a}=V_{m}+\left(1-x_{a}\right)\frac{\partial V_{m}}{\partial x_{a}}, \end{array}$$
(5c)$$\begin{array}{c} {\bar{V}}_{b}=V_{m}-x_{a}\frac{\partial V_{m}}{\partial x_{a}}. \end{array}$$

The relationship between volume $v_{m}$ (spatial, cm^3^, for example) and molar volume $V_{m}$ (Equations (Equation 4) or ([Disp-formula FD5a-sensors-24-04061]), cm^3^/mol) is given by $v_{m}=V_{m}n_{m}$. In the same way, partial molar volumes ${\bar{V}}_{a}$ and ${\bar{V}}_{b}$ allow the solution to be interpreted as a mixture, composed of substances a and b, which occupy spatial volumes $v_{a}={\bar{V}}_{a}n_{a}$ and $v_{b}={\bar{V}}_{b}n_{b}$, respectively. Since $n_{a}+n_{b}=n_{m}$ implies that $x_{b}=1-x_{a}$, this work proposes the variables $c_{a(x_{a})}$ and $c_{b(x_{a})}$ as functions of molar concentration $x_{a}$, which denote the propagation velocities in the partial molar volumes *a* or *b*, respectively, being formally different from the partial ultrasound speed defined by [18]. Thus, the layered model using the new set of variables becomes (Figure 1b Top):(6)$$c_{m}=\frac{V_{m}\cdot n_{m}}{\frac{{\bar{V}}_{a}n_{a}}{c_{a}\left(x_{a}\right)}+\frac{{\bar{V}}_{b}n_{b}}{c_{b}\left(x_{a}\right)}}=\frac{V_{m}}{\frac{{\bar{V}}_{a}x_{a}}{c_{a}\left(x_{a}\right)}+\frac{{\bar{V}}_{b}\left[1-x_{a}\right]}{c_{b}\left(x_{a}\right)}}.$$

The propagation velocity functions $c_{a(x_{a})}$ and $c_{b(x_{a})}$ have not been reported in the literature and there is no standard behavior for them. Even though functions based on polynomials of several degrees were tested in this work, a sixth-degree polynomial allowed an almost perfect correlation $R^{2}$ with the experimental data of the propagation velocity of the system, as will be shown in the Section 4. Other polynomials with degrees as fifth or fourth were tried; however, the correlation $R^{2}$ in those cases decreased significantly, being around 0.8. Highest-degree polynomials can be used without any significant fitting advantage as the seventh degree but with an increase in the number of coefficients. Then, a couple of new functions are proposed herein rather than constant values commonly used in the layered conventional (nonmiscible) approach. The fitting process (least square) is used to obtain the sixth-degree polynomials of the propagation velocity functions $c_{a(x_{a})}$ and $c_{b(x_{a})}$ given by:
(7a)$$\begin{array}{c} c_{a(x_{a})}=A_{a}x_{a}^{6}+B_{a}x_{a}^{5}+\ldots +G_{a}, \end{array}$$
(7b)$$\begin{array}{c} c_{b(x_{a})}=A_{b}x_{a}^{6}+B_{b}x_{a}^{5}+\ldots +G_{b}, \end{array}$$

The experimental propagation velocities $c_{m}$ are measured in the water molar fraction range from 0 to 1. Using the least squares algorithm, the constants $A_{a}$, $A_{b}$, $B_{a}$, $B_{b}$, … $G_{a}$ and $G_{b}$ are modified to adjust the theoretical and experimental curves described by Equation (Equation 6). The experimental data of $c_{m(xa)}$ measured in this work (see the Experimental Section), and the tabulated data of $V_{m}$, ${\bar{V}}_{a}$ and ${\bar{V}}_{b}$ from the literature [1] are used as input (see Figure 2).

The propagation velocity $c_{m}$ and the density $\rho_{m}$ of the mixture allow us to define its molar isentropic compressibility coefficient $K_{s}$ using the Newton–Laplace equation (conventional Equation ([Disp-formula FD8a-sensors-24-04061])) [7]. In this work, an alternative way to determine $K_{s}$ for miscible mixtures is proposed by entering $K_{sa}$ and $K_{sb}$ (see Figure 2) into an additive rule, which is associated with its molar composition and functions $c_{a(x_{a})}$ and $c_{b(x_{a})}$ (new Equation ([Disp-formula FD8b-sensors-24-04061])).
(8a)$$\begin{array}{c} K_{s}^{con}=\frac{1}{\rho_{m}c_{m}^{2}}, \end{array}$$
(8b)$$\begin{array}{c} K_{s}^{new}=\overset{K_{sa}}{\overset{︷}{\frac{1}{\rho_{a}c_{a(x_{a})}^{2}}}}x_{a}+\overset{K_{sb}}{\overset{︷}{\frac{1}{\rho_{b}c_{b(x_{a})}^{2}}}}\left[1-x_{a}\right], \end{array}$$
where $\rho_{a}=\frac{M_{a}}{{\bar{V}}_{a}}$ and $\rho_{b}=\frac{M_{b}}{{\bar{V}}_{b}}$ are calculated from partial molar volumes. Figure 2 shows the complete scheme algorithm, where the red and blue arrows denote inputs and outputs from the layered model (Equation (Equation 6)), respectively.

## 3. Materials and Methods

The measurement of propagation velocity in the mixture ($c_{m}$) was carried out using the in-house manufactured ultrasonic probe shown in Figure 3a. The probe consists of two square-radiating surface ultrasonic transducers and a working frequency of 3 MHz. The transducers were mounted on a metallic support to ensure good alignment and a fixed separation distance of 30 mm. The test is performed in both the transmission–reception mode and the pulse-echo mode (see Figure 1). The traveling time of the ultrasonic pulse between the emitter (Tx) and the receiver (Rx) is $t_{m}$. Rx receives the signal s(t) at time $t_{m}$ (transmission-reception mode) and Tx receives the signal q(t) at time $2t_{m}$ (pulse-echo mode). The traveling time is determined by cross-correlating the signals s(t) and q(t). From the traveling time and the known separation distance of the traducers, the propagation velocity can be established. In order to improve accuracy, the measurement system was calibrated using distilled water. Although applications of transducers of square radiating shape are not common, their performance is almost the same as the circular one in transient or pulse-echo mode, as used in this work. Subtle differences in the pattern of radiation are observed and can be relevant only in harmonic conditions when compared with the circular case [22,23].

The experimental setup is shown in Figure 3b. The test was performed in a thermostatic bath with a precision of 0.1 °C (CC-106A, Huber Kältemaschinenbau AG, Offenburg, Germany). An ultrasonic pulser/receiver (Olympus Panametrics model 5077-PR, Waltham, MA, USA) drives the ultrasonic probe. The pulser/receiver is connected to a digital oscilloscope (Agilent Technologies, model 5042, Santa Clara, CA, USA) to monitor and acquire ultrasonic signals. A digital thermometer (DeltaOHM, model HD2107.2, Caselle di Selvazzano (Padova), Italy), different from that included in the thermostatic bath, permanently monitors the temperature of the beaker content. The oscilloscope and the thermometer are connected to a desktop computer via LAN Network and USB, respectively. Specially developed Matlab scripts allow the simultaneous acquisition of the ultrasonic signals and temperature. In the case of the ultrasonic signals, both s(t) and q(t) were acquired using the two channels of the oscilloscope.

The test started with 100 mL of ethanol (Santa Cruz, 99.3° INPM) in a 1000 mL glass beaker. The beaker was partially immersed in the thermostatic bath for temperature control. The ultrasonic probe was immersed in the ethanol and the first ultrasonic signals were acquired. Next, using a burette (±0.06 mL), 5 mL of distilled water was added to the beaker, and the mixture was homogenized using a laboratory mixer at 100 rpm. The mixer (Fisatom, model 711, São Paulo, SP, Brazil) was turned off before each signal acquisition, but it was kept in the beaker throughout the experiment. The process of adding 5 mL of distilled water was repeated until a total of 40 mL was added. The process was repeated until 40 mL of water was added. To increase the molar fraction points ($x_{W}$), 20, 40, 50, 100, 150 and 300 mL of water were added to the solution of 100 mL of ethanol and 40 mL of water. Before each measurement, a waiting time of 5 min was established to ensure temperature homogenization (exothermic reaction). The complete experiment was repeated three times to obtain the mean and standard deviation values of the propagation velocity.

## 4. Results

The evaluation of the proposed model requires knowledge of the molar volume ($V_{m}$) and the partial molar volume of ethanol ${\bar{V}}_{E}$ and water ${\bar{V}}_{W}$. The data reported by [1] were used to perform a polynomial regression (Equation ([Disp-formula FD5a-sensors-24-04061])). The fitting coefficients are shown in Table 1. This polynomial expression (Equation ([Disp-formula FD5a-sensors-24-04061])) was used to evaluate molar and partial molar values (Equations ([Disp-formula FD5b-sensors-24-04061]) and ([Disp-formula FD5c-sensors-24-04061])) at the concentration values used in this work. These values are shown in Figure 4.

Figure 5 shows that both Rx and Tx signal a response pattern. Even though the pulse-receiver only applies one impulse, when a delta function is centered at 3 MHz, the transducers produce some ringing or reverberations due to the presence of matching layers. The cross-correlation between those signals allows us to determine the delay $t_{m}$ used to calculate the propagation velocity of the system. The cross-correlation is a well-studied and established technique to measure the delay in a set of successive signals [14,15].

Figure 6 shows a comparison of the experimental and theoretical propagation velocities of the mixture ($c_{m}$) as a function of the molar fraction of water. Along with the experimental results, the classical mixture model layers and the new proposed model were plotted. The experimental results show a maximum value of $c_{m}$ near $x_{W}=0.9$. Around this point, the values of $c_{m}$ are greater than the propagation velocity in both pure substances. These results are similar to those reported in the literature [7,24,25,26,27]. The new proposed method reproduces the experimental results with good accuracy. The percentage relative error is less than 1% at all evaluated points. On the other hand, the classical layered model provides a mixing propagation velocity that increases monotonically between $c_{E}$ and $c_{W}$. The behavior is parabolic and the peak value around $x_{W}=0.9$ is not modeled using the conventional approach.

The coefficients of the function $c_{m}$ (sixth-degree polynomial) generated by the new layers approach (solid line, Figure 6) are shown in Table 1. This experimental pattern can be described using the propagation velocity functions $c_{E}$ and $c_{W}$ shown in Figure 7, using the coefficients shown in Table 1. Although $c_{E}$ and $c_{W}$ show great variability over $x_{W}$, the values for the pure states are those expected, which means 1170 m/s at $x_{W}=0$ for $c_{E}$ and 1500 m/s at $x_{W}=1$ for $c_{W}$. In diluted states, the propagation velocity functions have values different from zero, even though the concentration of one of the compounds in the mixture is zero, i.e., $c_{W}$ = 1500 m/s at $x_{W}=0$ and $c_{E}$ = 2680 m/s in $x_{W}=1$.

The isentropic compressibility coefficient ($K_{s}$) was calculated using the functions $c_{W}$ and $c_{E}$ (Equations ([Disp-formula FD7a-sensors-24-04061]) and ([Disp-formula FD7b-sensors-24-04061]) and the coefficients from Table 1) and the density functions for each species in a simple mixture rule of Equation ([Disp-formula FD8b-sensors-24-04061]). The densities $\rho_{E}$ and $\rho_{W}$ were obtained from the partial molar volumes taken from [1] and the molecular weight. The isentropic compressibility coefficients of water ($K_{sW}$) and ethanol ($K_{sE}$) in the mixture are also functions of the volume fraction of water ($x_{W}$). The curves $K_{sW}$ and $K_{sE}$ are shown in Figure 8.

Figure 9 shows a comparison of $K_{s}$ as a function of the molar fraction of water calculated from the experimental propagation velocity in the mixture (Equation ([Disp-formula FD8a-sensors-24-04061])) and calculated from the new proposed model (Equation ([Disp-formula FD8b-sensors-24-04061])). The behavior is similar in both cases.

## 5. Discussion

The $K_{s}$ values obtained by the new approach to the layered model are smaller than those obtained by the classical approach and from $x_{w}=0$ to $x_{w}=0.9$, and the maximum deviation is 22% at $x_{W}=0.65$. A minimum value of $K_{s}$ occurs at high $x_{W}$ in both curves: at $x_{W}=0.9$ in the classical case and at $x_{W}=0.8$ in the modeled one. The minimum $K_{s}$ obtained by the classical approach is close to that reported in [7] at a temperature of 25 °C, using the same Equation ([Disp-formula FD8a-sensors-24-04061]) described here. In these results, impurities in the substances and differences less than 0.3% in the initial concentration of ethanol (in relation to the ethanol used by other authors) may be the main sources of error.

Finally, it should be noted that the functions $c_{W}$ and $c_{E}$ were initially proposed as inputs in the algorithm (Figure 2) to obtain the molar volumes $V_{m}$ and the partial molar volumes ${\bar{V}}_{E}$ and ${\bar{V}}_{m}$. This would be useful in chemically characterizing binary miscible systems using the propagation velocity data of the mixture. In this case, the curves of $c_{W}$ and $c_{E}$ were proposed from the Legendre transformations of the experimental propagation velocity solution (obtained by polynomial fitting, analogous to the molar volume). However, the molar volume and the partial volumes of the species calculated by the fitting process (coefficients *A*, *B*, *C* and *D* of Equation ([Disp-formula FD5a-sensors-24-04061]) into Equation (Equation 6) that minimize the error with the experimental data) disagree with the expected values. This means that the Legendre transformation only applies to conventional thermodynamic magnitudes, such as volume or enthalpy Michael Abbott [20].

## 6. Conclusions

A new approach to the layered model was implemented using partial molar volume and partial propagation functions. The results show a significant improvement in the description of miscible liquid systems. Although the model requires information from the same system that has already been tabulated, new physicochemical properties can be found. The propagation velocity and the isentropic compressibility coefficient for each chemical compound in the solution were estimated. The values were estimated on the basis of the measurement of the propagation velocity in the mixture, as well as on the tabulated data of the mixture molar volume and partial molar volume of each species. This technique has not yet been reported in the literature for miscible mixtures, and it can be further investigated for new applications other than water–ethanol systems.

## Figures and Tables

**Figure 1 sensors-24-04061-f001:**
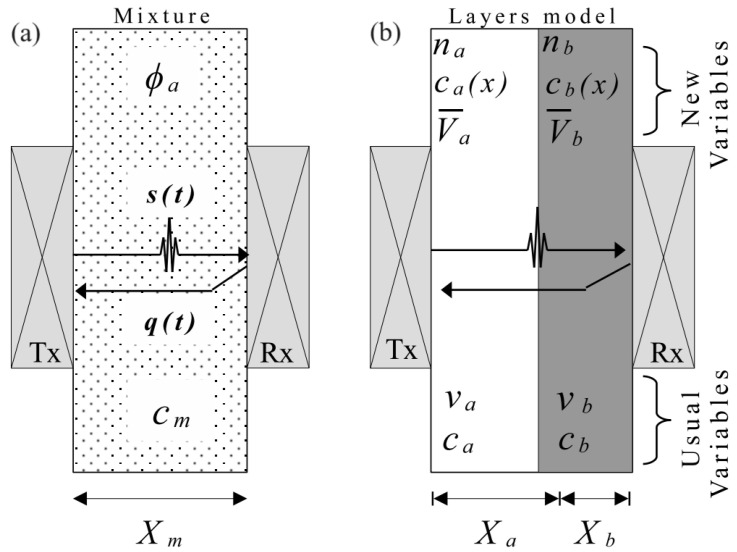
(**a**) Schematic diagram of pulse-echo and transmission techniques used to measure the propagation velocity of the mixture, (**b**) layered model considering the substances of the mixture as if they were separated in layers, taking into account the new approaching variables in the miscible case (top) and the conventional variables used in the nonmiscible system (bottom).

**Figure 2 sensors-24-04061-f002:**
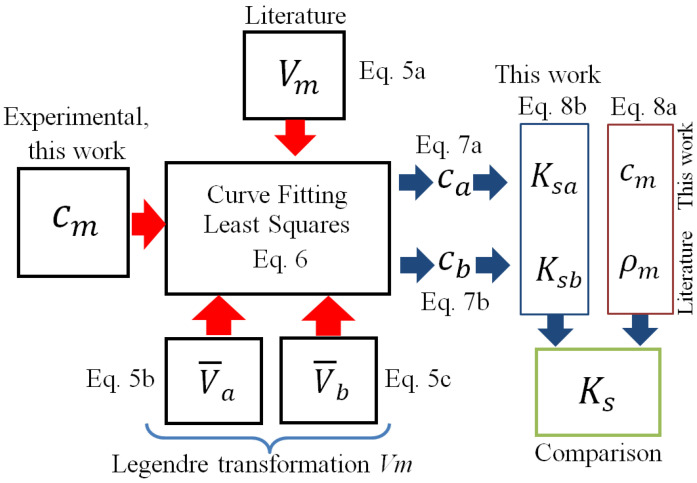
Scheme of experimental ($c_{m}$) and tabulated data ($V_{m}$, ${\bar{V}}_{a}$ and ${\bar{V}}_{b}$) input to calculate $c_{a(x_{a})}$ and $c_{b(x_{a})}$, which are used to determine $K_{sa}$ and $K_{sb}$. These coefficients are used to estimate the total $K_{s}$, which is compared to the conventionally calculated value.

**Figure 3 sensors-24-04061-f003:**
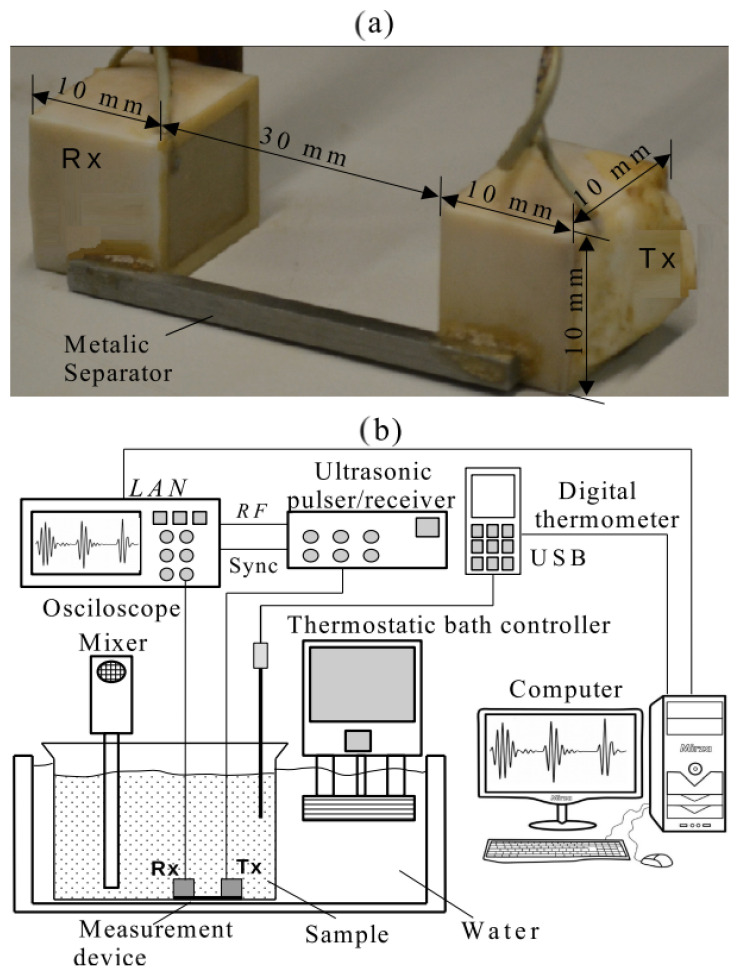
(**a**) Piezoceramic probe, (**b**) setup scheme.

**Figure 4 sensors-24-04061-f004:**
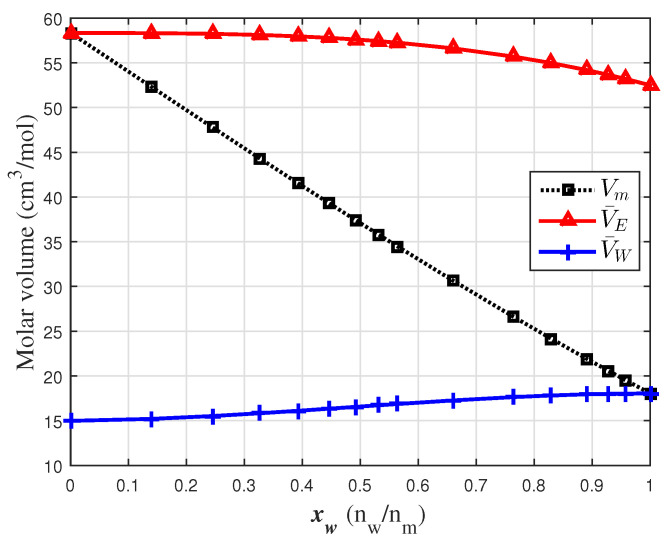
Water–ethanol molar volume properties at 20 °C [1].

**Figure 5 sensors-24-04061-f005:**
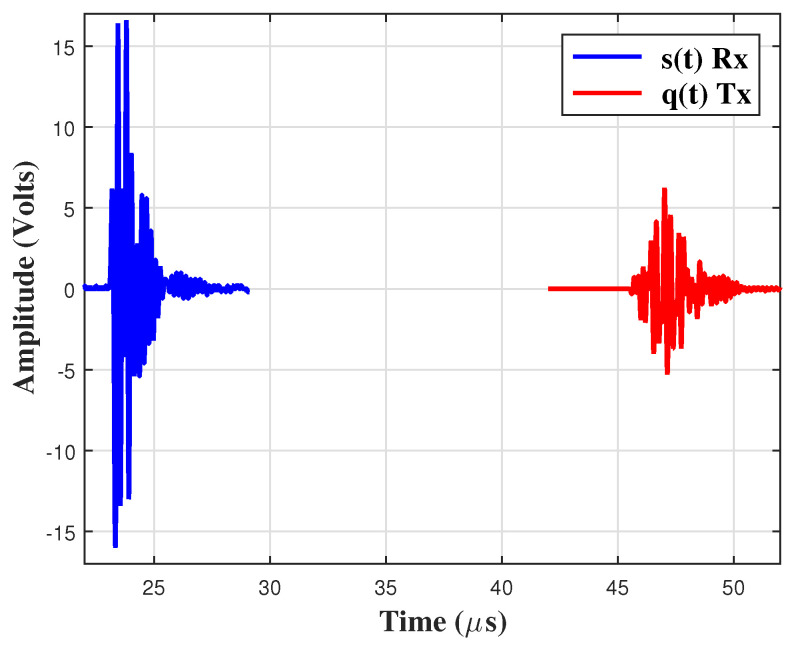
Response pattern at 20 °C and $x_{w}=0$.

**Figure 6 sensors-24-04061-f006:**
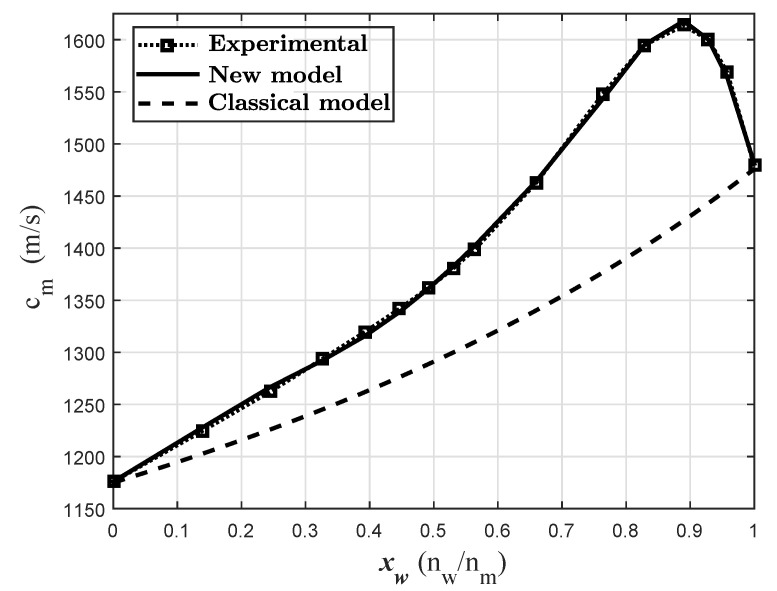
Experimental propagation velocity: new and conventional layers approach comparison at 20 °C.

**Figure 7 sensors-24-04061-f007:**
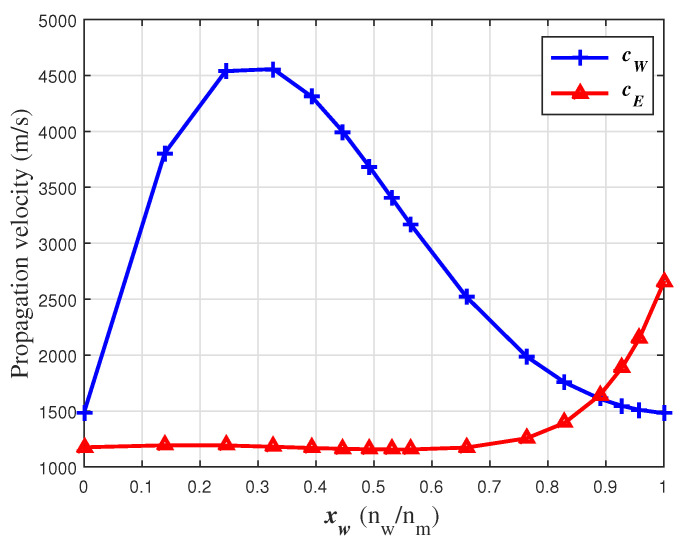
Water and ethanol propagation velocity functions at 20 °C.

**Figure 8 sensors-24-04061-f008:**
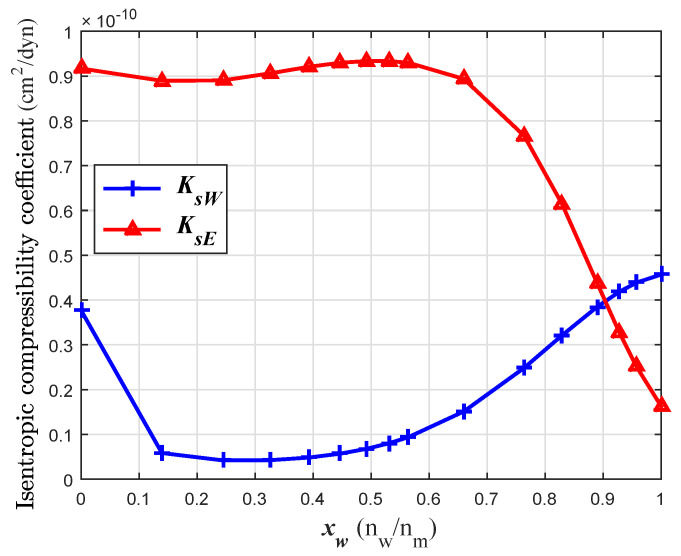
$K_{sW}$ and $K_{sE}$ functions in the water–ethanol solution at 20 °C.

**Figure 9 sensors-24-04061-f009:**
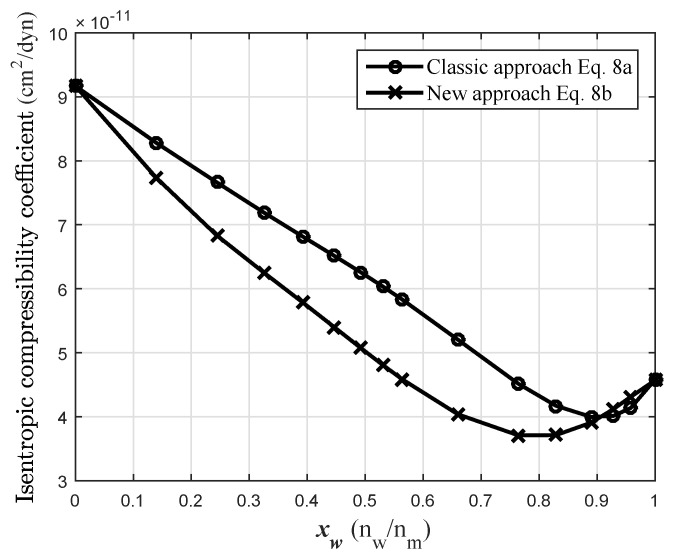
Total $K_{s}$ comparison to the solution water–ethanol at 20 °C, $dev_{max}=22\%$ ($x_{W}=0.65$).

**Table 1 sensors-24-04061-t001:** Coefficient of polynomial functions.

Coeff.	$V_{m}$ (cm^3^/mol)	$c_{m}$ (m/s)	$c_{E}$ (m/s)	$c_{W}$ (m/s)
*A*	2.7815	$-1.17\times 10^{4}$	$2.52\times 10^{4}$	$4.55\times 10^{4}$
*B*	0.2814	$2.58\times 10^{4}$	$-5.59\times 10^{4}$	$-1.67\times 10^{5}$
*C*	−43.352	$-2.06\times 10^{4}$	$4.86\times 10^{4}$	$2.16\times 10^{5}$
*D*	58.334	$7.74\times 10^{3}$	$-1.92\times 10^{4}$	$-9.14\times 10^{4}$
*E*	-	$-1.29\times 10^{3}$	$2.79\times 10^{3}$	$-2.43\times 10^{4}$
*F*	-	$4.27\times 10^{2}$	−0.01	$-2.13\times 10^{4}$
*G*	-	$1.17\times 10^{3}$	$1.17\times 10^{3}$	$1.487\times 10^{3}$

## Data Availability

Data are contained within the article.

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
