# Peer review of "Determination of Partial Propagation Velocity and Partial Isentropic Compressibility Coefficient in Water–Ethanol System"

_sensors, 2024, doi:10.3390/s24134061_

Round 1

Reviewer 1 Report

Comments and Suggestions for Authors

In this paper, the propagation velocity functions for each substance in the solution were calculated by using the partial molar properties in the layers model. These results were used to determine the isentropic compressibility coefficient (ICC) for each species in the water-ethanol system. Finally, the total value of ICC was estimated using an additive mixing rule and compared to value, calculated conventionally.

Some shortcomings of the paper are the following:

1. (Line 124): 2tm must be replaced by 2tm

2. Figs. 7, 8: At the ordinate axis, it should be pointed: “isentropic compressibility coefficient”.

3. Fig. 8: What the devmax means?

4. All references, relating to papers, must be accompanied by doi.

Author Response

Thank you for your comments. All comments and suggestions have been carefully addressed in the attached file.

Reviewer 2 Report

Comments and Suggestions for Authors

The authors proposed a novel approach to determine isentropic compressibility coefficient for each component of the binary mixture using layered model and derived analytical relations. The latter is used to compute the propagation velocity and the coefficient for each chemical compound using ultrasonic measurements with two probes. A good agreement with experimental data is demonstrated. The paper is of good quality and might be recommended for publication after major revisions. The comments are given below:

- Layered model is more suitable term than layers model, and it is suggested to provide the substitution.

- Please explain for the readers why do you apply Legendre transformation to Eq. (5a).

- Why third degree polynomial equation was utilized? The choice must be explained.

- Sixth degree polynomials choice for the propagation velocity functions c were also not explained. Is it caused by the relation (6) and the previous choice of the interploation polynomials for Vm?

- K_{s−con} and K_{s−new} seems to be complicative, K_{s}^{con} or another notation would be better. 

- The authors do not provide any details related to the ultrasonic signal employed. Is it sine pulse?

- More details on square-radiating surface ultrasonic transducers must be provided.

Author Response

(The authors gave the same response as above.)

Round 2

Reviewer 2 Report

Comments and Suggestions for Authors

The authors have sufficiently corrected the paper, so I recommend it for publication.